



# Spatial prediction of earthquake-induced landslide probability

Robert N. Parker[1], Nicholas J. Rosser[2], Tristram C. Hales[1]

[1] School of Earth and Ocean Sciences, Sustainable Places Research Institute, Cardiff University

[2] Institute of Hazard, Risk and Resilience, Durham University

*Correspondence to*: Robert N. Parker (parkerr5@cardiff.ac.uk)

**Abstract.** We developed a generalized model to describe and predict the spatial distribution of earthquake-induced landslides, based on a regression analysis of 9 co-seismic landslide inventories from different earthquakes and regions. Our model expresses the absolute spatial probability of landslides as a function of peak ground acceleration and hillslope gradient, based on data from global topographic and seismic ground motion datasets. The output from our model predicts

probabilities for landslides triggered in sedimentary, meta-sedimentary, igneous and volcanic lithology, and is applicable to shallow continental earthquakes of moment magnitude range 6.2 to 7.9, and depths between 10 and 21 km. To obtain absolute probability predictions, we use only landslide source areas as input data, and explicitly estimate and correct for known incompleteness in input datasets, through a novel Monte Carlo approach. We estimate the uncertainty of these predictions, through extensive testing of the performance of the model, when making out-of-sample predictions for all 9

earthquakes. Our model is notably simpler than others developed to predict spatial probability of landsliding, as we have only included variables that could be constrained consistently at the global-scale, and eliminated those that did not influence landslide probability in a consistent manner across all earthquakes in our dataset. The model outputs also provide a baseline to further investigate spatial and temporal sources of unexplained variability in co-seismic landslide distributions. Using freely available topographic and ground motion data, we suggest that our model can be applied more widely, to provide

landslide predictions for earthquakes with no landslide data.

## 1 Introduction

Large earthquakes in steep mountainous regions initiate a spatially and temporally complex sequence of landsliding commonly across extensive areas (Keefer, 2002, Bommer and Rodriguez, 2002, Rodríguez et al., 1999). These include landslides that occur during or shortly following seismic ground motions (co-seismic landslides), and landslides triggered by

aftershocks and rainfall on seismically destabilised hillslopes (post-seismic landslides, e.g.: Marc et al. (2015), Tang et al. (2011)). Increasingly, landslides triggered by earthquakes have been recorded in landslide inventory datasets, providing information on hundreds of thousands of earthquake-induced slope failures (Marc et al., 2016, Keefer, 2002, Rodríguez et



al., 1999, Marc et al., 2017). In this investigation, we analyse data from 9 major earthquakes that triggered landslides, to develop a generalized model for predicting the spatial probability of co-seismic landslides.

The probability of landslide occurrence during an earthquake can be expressed as a function of factors that influence the dynamic response of hillslopes subject to seismic shaking (e.g.: Jibson, 2011, Newmark, 1965). These factors can be broadly grouped into those that influence the intensity of event-specific ground motions, and the strength of hillslope materials and the static shear stresses acting upon them. Empirical studies have revealed several proxy variables that can be used to represent these factors at the regional scale (Table 1). Treating landslide occurrence as a conditional probability problem, supervised learning models have been used to describe the spatial distribution of landslide probability as a function of various subsets of these variables (Korup and Stolle, 2014). Most previous work has focussed on analysis of landslides specific to a single earthquake or region (Yilmaz and Keskin, 2009, Lee et al., 2008, Kamp et al., 2008, Garcia-Rodriguez et al., 2008, Lee and Evangelista, 2006, Ayalew and Yamagishi, 2005, Lin and Tung, 2004). Models based on these studies are necessarily regionally specific, lacking a sufficient spread of conditions to make them transferable between events. From compilations of data from multiple earthquakes, the landslide response to ground motions has been expressed as a function of event-scale summary variables, representing ground motion and topographic conditions across the landscape (Marc et al., 2016, Keefer, 1984, Malamud et al., 2004b). This work has provided a seismologically consistent expression to predict the total area, volume and extent of co-seismic landslides, applicable globally (Marc et al., 2017, Marc et al., 2016). However, such approaches lack detail on the spatial distribution of landslides, which is arguably an important output if such models are to be used to assess earthquake impacts.

Methods using data from multiple earthquakes, to derive statistical models generalized at 1 km$^2$ resolution (Nowicki et al., 2014), and 90 m resolution (Kritikos et al., 2015), have had some success. These models perform well in identifying the likely locations of landslides in the landscape, based upon a relatively small number of earthquakes. Challenges remain around the relatively small number of landslide inventories used to train and test these models, making it difficult to assess their wider utility, and the value of the relative rather than absolute probabilities that they generate. A significant challenge with such models is also how they deal with small landslides, which whilst commonly the most numerous, are those most difficult to consistently map. As a result, landslide inventories are commonly incomplete, due to censoring of small landslides to various degrees (e.g.: Guzzetti et al., 2002a, Malamud et al., 2004a). Many inventories also do not differentiate landslide source and runout areas within the mapped landslide footprint. This distinction is however vital as models aim to predict the zone of failure rather than the full landslide extent.

Physically-based models of regional earthquake-triggered landslides commonly employ a form of the Newmark sliding block model, which estimates seismically-driven displacements in hillslopes (Jibson, 2011, Godt et al., 2009). Where rock-strength parameters and displacement parameters for failure are known, it is possible to derive relative spatial and absolute



likelihoods of landsliding (Gallen et al., 2015, Gallen et al., 2016). However, poor constraints on these parameters from regional-scales and above, limit the extent to which such physically-based models can be reliably applied more widely.

To develop a model applicable to a broad range of earthquakes, we take a statistical, conditional probability approach. First,
5   we apply a systematic and automated technique to approximate the separation between landslide source and runout zones. Next, we undertake an analysis of the landslide-area frequency distribution, to assess and correct for the relative completeness of landslide datasets due to (possible) differences in mapping. Our modelling methodology is designed to avoid overfitting, stipulating that any relationship included in our model must be consistently present in data from all inventories in our dataset. In this way, we develop a model that is simplistic in terms of its predictor variables, widely
10   applicable to earthquakes of the types included in our sample, within quantified bounds of uncertainty. Where necessary we make analytical decisions likely to yield a conservative model. For example, we favour the over-estimation of landslide probability, considering uncertainties and potential sources of error in our analysis, which are noted explicitly throughout the methodology.



**Table 1. Summary of proxy variables suggested to influence spatial distributions of earthquake-induced landslides, based on empirical studies**

| Proxy variable | Mechanical link to landslide occurrence | Case studies |
|---|---|---|
| **Seismic forcing (Ground motion intensity)** | | |
| Seismic wave attributes (e.g., PGA, PGV, PGD, Arias Intensity, MMI) | Local metric of shaking intensity | Meunier et al. (2007), Dai et al. (2011), Lee et al. (2008), Meunier et al. (2013),Hancox et al. (2002), Hancox et al. (1997) |
| Distance from the seismic source | Regional attenuation of seismic wave amplitudes | |
| Position on hillslope (normalised distance from stream to ridge crest, and topographic position index) | Ridge to stream patterns of topographic amplification and damping | Davis and West (1973), Bouchon (1973), Wu et al. (1990), Benites et al. (1994), Meunier et al. (2008), Densmore et al. (1997), (Kritikos et al., 2015) |
| Orientation of hillslope relative to seismic source | Directional patterns of topographic amplification and damping, due to the incidence angle of seismic waves | |
| Hanging wall vs. footwall location of sites | Proximity of the fault and enhanced rupture directivity effects in hanging wall areas | Abrahamson and Somerville (1996), Somerville et al. (1997), Abrahamson et al. (2008) |
| **Strength of hillslope materials** | | |
| Bedrock lithology | Hillslope material strength | Khazai and Sitar (2004), Parise and Jibson (2000), Keefer (2000), Dai et al. (2011) |
| Structural geology (discontinuities) | Kinematic feasibility, i.e. orientation of bedrock discontinuities relative to slope aspect and topography | Hoek et al. (2002), Selby (2005), Moore et al. (2009) |
| Northerly component of hillslope aspect | Relative intensity of rock breakdown via physical and chemical weathering | Meunier et al. (2008), Parker (2010), Chen et al. (2012), Parker (2013), Mcfadden et al. (2005) |
| Rainfall | The effect of pore water pressure in reducing hillslope effective stress | (Dellow and Hancox, 2006, Iverson, 2000) |
| **Static stress loading in hillslopes** | | |
| Hillslope gradient | Magnitude of static stress loading in hillslopes | Keefer (2000), Khazai and Sitar (2004), Lee et al. (2008), Dai et al. (2011) |
| Local hillslope relief | | |



### 1.1 Landslide datasets

We use 9 landslide inventories from previous studies (Table 2). These include landslides triggered by earthquakes in the moment magnitude ($M_w$) range 6.2 - 9 and focal depth range 10 - 29 km. The locations of the earthquakes span a range of climatic conditions including tropical, sub-tropical, cool sub-tropical, dry Mediterranean, montane temperate, and oceanic

temperate. The study regions span a broad range of geological units and rock types, from carbonates to volcanic rocks and metasediments. In each inventory, co-seismic landslides have been mapped from aerial or satellite imagery that was acquired within 1-30 days of the mainshock, combined with varying levels of field reconnaissance and validation. Mapping scales range from 1:1000 to 1:60000 (equivalent to 0.5-30 m image pixel resolution (Tobler, 1987, Tobler, 1988)). Landslide polygons represent ground disturbed by landslides, that includes the scar, runout and deposit zones, but excludes debris-flow

transport and deposition along pre-existing channels. This incorporates a range of landslide types, which most inventories do not explicitly differentiate between. We focus on landslides that involve the collapse of hillslopes, where the failed mass typically evacuates the failure scar and moves downslope to leave a discernible, bare-earth scar. As per the classification of (Keefer, 1984, Keefer, 2002), these types of failures generally fall into the categories of *disrupted slides* and *coherent slides*. Most landslide inventories are dominated by disrupted and coherent slides, which account for around 94% of reported

earthquake-induced landslides (Keefer, 1984, 2002, Mcfadden et al., 2005).

The spatial completeness of these datasets is likely to vary greatly, depending on the availability of imagery and quality of mapping. For example, landslide data from the 2014 Ludian earthquake covers only a small proportion of the total area of steep terrain that experienced strong ground motions. For this reason, in our modelling we use a conditional probability

approach to analyse the occurrence of landslides, which considers at-a-point triggering conditions under which hillslopes did and did not fail, rather than considering the wider event-scale co-seismic landslide response. Definition of the areal footprint of landslide mapping is an important consideration, as this defines the limit of where hillslopes did not fail. Where the image footprint of landslide mapping was known in the case of the 2008 Wenchuan earthquake, this was used to define this limit. In the absence of image footprints for other events, the limit was approximated by a convex hull polygon bounding the limits

of the landslide spatial distribution. This allows us to assimilate spatially incomplete landslide inventories without biasing our analysis.

To analyse the spatial distribution of co-seismic landslides, we model landslide source areas. It is typically challenging to visually separate landslide source and runout-deposit areas when mapping from aerial and satellite imagery. None of the

landslide inventories used here incorporate this distinction. We systematically differentiate source from runout-deposit area by dividing the landslide footprint at its midpoint elevation(Parker et al., 2015). Although this approach is similar to the method of extracting landslide areas above the median landslide elevation (e.g.: Parise and Jibson, 2000, Jibson et al., 2000, Refice and Capolongo, 2002, Lee et al., 2012), our technique is less prone to overestimation of the source area for landslide

masses that runout over large distances across low gradient terrain. Comparison of results from this technique with source and runout zones separated visually in high-resolution imagery, suggests that this provides a good approximation of the separation between landslide source and runout-deposit zones (Parker et al., 2015)

5   Amalgamation of multiple landslide features into single mapped polygons is problematic when the characteristics of individual landslides are pertinent, such as deriving volumetric estimates through volume-area scaling relationships for individual landslides (Marc and Hovius, 2015). Our technique for extracting landslide source areas is insensitive to all but the most severe amalgamation effects, where landslide polygons link across and along valley bottoms, spanning an unrealistic elevation ranges. To minimise this, all landslide inventories used have been checked and corrected for any severe

10  amalgamation effects (Marc et al., 2016, Marc and Hovius, 2015, Li et al., 2014, Wartman et al., 2013). As our model is also based on predicting the areal coverage of landslide source zones, rather than the characteristics of individual landslides, this sensitivity is also of lesser importance.



**Table 2. Earthquake events and physiographic settings**

| Event name | Latitude | Longitude | Magnitude, Mw | Focal Depth, km | Strike | Dip | Rake | Main Lithology | Climatic setting |
|---|---|---|---|---|---|---|---|---|---|
| 1983 Coalinga | 36.232 | -120.312 | 6.1 | 10.0 | - | - | - | Sedimentary | Dry Mediterranean |
| 1994 Northridge | 34.213 | -118.537 | 6.7 | 19.0 | 121 | 46 | 102 | Sedimentary | Dry Mediterranean |
| 1999 Chi-Chi | 23.772 | 120.982 | 7.7 | 21.0 | 209 | 68 | 89 | Metasedimetary | Sub-Tropical |
| 2004 Niigata | 37.226 | 138.779 | 6.6 | 16.0 | 211 | 53 | 92 | Sedimentary / Volcaniclastics | Oceanic Temperate |
| 2008 Iwate | 39.03 | 140.881 | 6.9 | 10.0 | 28 | 40 | 100 | Volcanoclastics / Igneous | Oceanic Temperate |
| 2008 Wenchuan | 31.002 | 103.322 | 7.9 | 19.0 | 337 | 76 | 63 | Metasedimetary / Igneous | Cool Sub Tropical |
| 2010 Haiti | 18.443 | -72.571 | 7.0 | 13.0 | 249 | 72 | 8 | Sedimentary / Limestone | Tropical |
| 2011 Honshu | 38.297 | 142.373 | 9.1 | 29.0 | 192 | 8 | 78 | Sedimentary / Volcaniclastics / Igneous | Oceanic Temperate |
| 2014 Ludian | 27.190 | 103.409 | 6.2 | 12.0 | 338 | 74 | -32 | Metasedimetary / Igneous | Cool Sub Tropical |





**Table 3. Landslide inventory information**

| Event name | Number of mapped landslides | Total landslide area (km²) | Total landslide source area (km²) | $A_{min}$ | α | Inventory percentage completeness where landslide source area > $10^3$ km² | Total mapped extent area (km²) | Mapping scale | Mapping resolution (m) | Mapping imagery lag period | Landslide data source |
|---|---|---|---|---|---|---|---|---|---|---|---|
| 1983 Coalinga | 3980 | 4.75 | 2.16 | 869 | 2.60 | 100 | 1537 | 1:24000 | 12 | 1 day | Harp et al. 1990 (edited by Marc et al. 2016) |
| 1994 Northridge | 11111 | 23.83 | 11.41 | 6149 | 3.19 | 38+/-8.7 | 4043 | 1:60000 | 30 | 8 hours | Jibson et al. 1994 (edited by Marc et al. 2016) |
| 1999 Chi-Chi | 9272 | 127.57 | 53.98 | 7687 | 2.30 | 63+/-2.3 | 10499 | 1:20000 | 10 | 6 days | Liao & Lee 2000 (edited by Marc et al. 2016) |
| 2004 Niigata | 10526 | 11.98 | 6.01 | 325 | 2.22 | 100 | 817 | 1:10000 | 5 | <30 days | Yagi et al., 2007 (edited by Marc et al. 2016) |
| 2008 Iwate | 3533 | 14.00 | 5.62 | 520 | 1.87 | 100 | 666 | 1:10000 | 5 | <30 days | Yagi et al., 2009 (edited by Marc et al. 2016) |
| 2008 Wenchuan | 57402 | 396.23 | 197.10 | 23996 | 3.14 | 13+/-2.4 | 37508 | 1:20000 | 0.5-10 | <30 days | Li et al. 2013 |
| 2010 Haiti | 23567 | 24.86 | 11.52 | 1197 | 2.28 | 98+/-0.4 | 3750 | 1:1200 | 0.6 | 16 days | Harp et al. 2016 (edited by Marc et al. 2016) |
| 2011 Honshu | 2203 | 1.56 | 0.73 | 1209 | 3.20 | 93+/-3.2 | 37944 | 1:1000 | 0.5-2.5 | 1-25 days | Wartman et al. 2013 |
| 2014 Ludian | 1024 | 5.19 | 2.16 | 4210 | 2.54 | 65+/-9.0 | 232 | 1:20000 | 2-10 | 3-4 days | Xu 2015 (edited by Marc et al. 2016) |





### 1.2 Assessing landslide inventory completeness

An inverse non-linear relationship between size and frequency is observed for landslides (e.g.: Pelletier et al., 1997, Hovius et al., 1997, Hovius et al., 2000, Dai and Lee, 2001, Guzzetti et al., 2002b, Malamud et al., 2004a, Malamud et al., 2004b, Van Den Eeckhaut et al., 2007). This distribution has been associated with self-organised criticality in the hillslope failure

process (Turcotte and Malamud, 2004, Hergarten and Neugebauer, 2000), and the mechanics of landslide rupture and the geometry and abundance of discontinuities in soil and rock (Brunetti et al., 2009, Stark and Guzzetti, 2009). For small landslides, many inventories exhibit a reduction in the frequency density relative to power-law scaling, commonly termed the 'rollover'.

For complete landslide inventories, this rollover can be understood as a physical manifestation of the conditions of hillslope failure (Pelletier et al., 1997, Van Den Eeckhaut et al., 2007), controlled by the availability of hillslopes and transition from cohesion controlled failure in shallow landslides to friction controlled failure in deep-seated landslides (Guzzetti et al., 2002b, Stark and Guzzetti, 2009, Frattini and Crosta, 2013). In many datasets, the rollover occurs at higher areas or volumes due to censoring of smaller landslides resulting from the mapping technique (Hovius et al., 1997, Hovius et al., 2000, Stark

and Hovius, 2001, Brardinoni and Church, 2004). The position of the rollover can therefore also be interpreted as indicating the limit of landslide size below which the datasets become incomplete (Guzzetti et al., 2002b, Malamud et al., 2004b, Malamud et al., 2004a). This incompleteness results in an under-estimation of the occurrence of and hazard contributed by smaller landslides. The severity of censorship of smaller landslides varies for landslide inventories mapped from imagery of different resolutions. This in turn results in an inconsistent level of under-estimation of landslide probability and hazard from

one dataset to the next. To correct for censoring, we approximate the completeness of each landslide inventory through analysis of the area-frequency distribution of landslide source zones (Brunetti et al., 2009). Using Alstott et al. (2014), we fitted power-law parameters for each dataset using the function:

(1)

$$p(A) = \frac{\alpha - 1}{A_{min}} \left( \frac{A}{A_{min}} \right)^{-\alpha}$$

where $p(A)$ is the probability of a landslide having a given area $A$, $A_{min}$ is the minimum size of landslide modelled by the function and $\alpha$ is the power-law scaling exponent. After Clauset et al. (2009), power-laws are fitted using the method of maximum likelihood. $A_{min}$ is determined by creating a power-law fit starting from each unique value in the dataset, then selecting the fit that results in the minimal Kolmogorov-Smirnov distance, between the data and the fit (Figure 1).






Best fit $\alpha$ parameters exhibit variability within the range of $\alpha$ values reported globally (Van Den Eeckhaut et al., 2007). $A_{min}$ values range from 325 m$^2$ to 23996 m$^2$ (Table 3). This suggests a large range of variability in the completeness of the landslide inventories. To correct for the apparent censorship of small landslides, we estimate the total area of missing landslides through extrapolation of the power-law fit for each dataset, to a target $A_{min\_target}$ value of 10$^3$ m$^2$. This

represents a landslide size larger than $A_{min}$ values observed in the most complete datasets for the Niigata, Iwate and Coalinga earthquakes. By choosing this value, we assume that, were other datasets similarly complete, departure from power-law scaling would occur at areas smaller than 10$^3$ m$^2$. We estimate the likely area of censored landslides, between $A_{min}$ for each dataset and our target $A_{\mathrm{min\_target}}$ value, using Monto Carlo simulations based on the fitted power-law functions and their uncertainty. For each dataset, we randomly generated 1000 synthetic populations of landslide areas from

the fitted power-law function, selecting $\propto$ values randomly within the range of estimated uncertainty. For each population, we then randomly sampled synthetic landslide areas, until the total area of synthetic landslides larger than $A_{min}$ was equal to that for the observed population of landslides (See: Appendix 1). The mean total area of landslides sampled provides an estimate of the corrected total landslide area for each earthquake inventory (Table 3).

The 2010 Haiti (98 ± 0.4% complete) and 2011 Honshu (93 ± 3.2% complete) datasets exhibit slight censoring, suggesting that the total landslide area captured by the datasets was incomplete. Potential censoring effects are larger for 2014 Ludian (65±9.0% complete), 1999 Chi-Chi (63±2.3% complete), 1994 Northridge (38±8.7% complete), and 2008 Wenchuan (13±2.4% complete). Although the estimated censoring in some datasets is high, the model outputs are supported in instances where landslides have been remapped at higher resolution. For example, Figure 2 shows comparative landslide

mapping from the 2008 Wenchuan earthquake over the 51 km$^2$ Taoguan basin, where a very high density of landslides was triggered close to the epicentre. For landslide source areas larger than 10$^3$ m$^2$, the full-event landslide inventory used in this investigation (Li et al. 2014), shows 254 landslides with a total source area of 1.2 km$^2$, while the high-resolution landslide inventory shows 2184 landslides with a total source area of 4.3 km$^2$. For this small area (<0.002% of the total mapped area of landslides triggered by the earthquake), the Li et al. (2014) dataset is 28% complete, suggesting that 13±2.4%

completeness for the whole event coverage may not be unreasonable.

## 2 Generalised statistical model of landslide spatial probability

Parametric and non-parametric supervised learning techniques have been used to analyse the spatial distribution of landslides (see Korup and Stolle, 2014), by modelling the influence of multiple predictor variables on a categorical response. Functions take the form:

30    (2)

$$P(Y = 1) = f(x_1, x_2, x_3 \dots x_n)$$



where the probability that $Y = 1$ is estimated given the values of one or more predictor variables ($x, x_n$ ...). In this case, Y = 1 corresponds to the occurrence of a landslide at a particular location. Logistic regression is the most popular parametric technique for assessing the controls on earthquake-triggered landslide distributions (e.g.: Yesilnacar and Topal, 2005, Dai

and Lee, 2003, Garcia-Rodriguez et al., 2008, von Ruette et al., 2011), favoured for its ease of implementation, interpretable parameters and predictive performance. An important consideration of logistic regression is that it can only characterize monotonic relationships between predictor and response variables, unless non-linear transforms are applied to the input data. Non-parametric approaches utilising Naïve Bayes, Artificial Neural Networks and Decision Tree (Random Forest) techniques have also been applied effectively in landslide modelling (Tsangaratos and Ilia, 2016, Youssef et al., 2016,

Ermini et al., 2005), which are essentially model free, i.e.: not restricted to particular mathematical functions. During exploratory data analysis we found a tendency of non-parametric techniques to produce over-fitted models, with relatively poor predictive performance as compared to logistic regression. This is because relationships between the probability of hillslope failure ($P_{LS}$) and the predictor variables we use are predominantly monotonic and well-approximated by the logistic function. For this reason, we use logistic regression as the basis for our analysis and predictive model building. The logistic

regression function takes the form:

(3)

$$P(Y = 1) = \frac{1}{1 + exp\left(-(b_0 + b_1x_1 + b_2x_2 + b_3x_3 \ldots b_nx_n)\right)}$$

where logistic regression is used to estimate the coefficients ($b, b_n$ ...) for predicting the probability that $Y = 1$, given the

values of one or more predictor variables ($x, x_n$ ...). In this case, Y = 1 corresponds to the occurrence of a landslide source area at a particular location.

We analyse the spatial distribution of landslides using variables that can be obtained consistently using publically available global datasets. Topographic data for this analysis is provided from the SRTM 1 arc second (30 m) global elevation dataset

(v 4.2, (NASA JPL, 2013). This data was collected in February 2000 (NASA JPL, 2013), after the Coalinga, Northridge and Chi-Chi earthquakes, but prior to the Niigata, Iwate, Wenchuan, Haiti and Ludian earthquakes. For the three earlier events there is the possibility that hillslope gradients measured at landslide sites may not accurately reflect slope characteristics at the time of the earthquake. However, as landslides are commonly translational and the estimated depths of over 99.99% of mapped landslides are less than uncertainties in the elevation data, most surface changes produce by the Coalinga,

Northridge and Chi-Chi earthquakes are unlikely to be detectable in the elevation model (See Appendix 3). As our analysis considered only the source area of landslides, any bias that this invokes is likely to involve the overestimation of hillslope gradients, in source areas where landslide head scarps have been steepened, or no effect in cases of translational landslides. This could give the appearance of landslide triggering on steeper slopes during post-2000 earthquakes than for pre-2000





earthquakes, with the effect revealed in systematically lower hillslope gradient regression coefficients for later events. As we do not see this effect, the use of post- landslide elevation data does not appear to affect the outcome of our analysis.

The USGS Shakemap Atlas (Allen et al., 2008) maps peak ground motions and intensity, termed "ShakeMaps", for recent

and historical earthquakes. Each map has been produced using the Shakemap methodology (Wald et al., 1999, Wald et al., 2005), with constraints from macroseismic intensity data, instrumental ground motions, regional topographically-based site amplifications, and published earthquake-rupture models. In general, ground motions are well-constrained near seismic recording stations and uncertainty increases with distance from seismic stations where ground motion prediction equations must be relied upon (Wald et al., 2008). Gridded ground motion and uncertainty data are made available at 900 m resolution.

We base our analysis on ground motion data rather than a distance-from-seismic-rupture proxy variable (e.g.: Lee et al., 2008), given the closer mechanical link to landslide triggering and the greater availability of ground motion data as compared to rupture models. Rock type and associated variability in material properties plays an important role in hillslope response to seismic accelerations (Parise and Jibson, 2000). In the absence of global lithological data mapped consistently at scale relevant to hillslopes, we are not able to incorporate lithology or rock type differences explicitly into our current

analysis. However, given feedbacks between rockmass strength properties and topographic slope and relief (e.g.: Schmidt and Montgomery, 1995), it is likely that some effects of lithology are implicitly incorporated.

To compare the spatial occurrence of landslides with these predictor variables, we defined a sample grid at 30 m resolution, based on the resolution of the digital elevation model. Response and predictor variables for analysis were then generated for

each grid cell. For the response variable, binary grids of landslide and non-landslide pixels were generated from the mapped landslide inventories. Pixels were classified based on whether any part of the pixel was touched by a landslide polygon (Y=1) or not (Y=0). In this way, our characterisation of the total area of hillslopes affected by landslides is again conservative, favouring over-estimation of landslide probability. For each grid cell, ground motion variables were sampled from Shakemap gridded data, incorporating Modified Mercalli Intensity (MMI), peak ground acceleration (PGA), peak

ground velocity (PGV) and peak spectral acceleration at 0.3s, 1s and 3s periods (PSA) and standard deviation of peak ground acceleration (SDPGA). Local hillslope gradient was measured over the DEM using a 3-pixel (90 m) window. Local relief (elevation range) and topographic position (pixel elevation divided by mean local elevation) were measured for radii of 30 to 500 m, to consider a range of local hillslope length scales. Upslope contributing area was calculated using discretised 8-direction (D8) and diffusive (D-infinity) flow routing techniques (Sharp, 2016). In the absence of a systematic technique for

defining channel heads from 30 m topographic data and across a range of climatic conditions (Clubb et al., 2014), we omit variables requiring derivation of stream channels, which could not be practically used in a predictive application. We also omit variables concerned primarily with characterising the aspect of hillslopes due the current lack of consensus on the exact control of slope aspect on co-seismic landslides (e.g.: Parker et al., 2015, Meunier et al., 2008, Xu et al., 2014).



We combined data from all 10 earthquakes to generate a single combined dataset for analysis. To correct for the effect of small landslide censoring on aggregate landslide probabilities in our analysis, the number of non-landslide samples selected from each dataset ($N_{NLS}$) was set according to the ratio of the total observed (uncorrected) landslide area ($A_{LS}$) to the total estimated (corrected) landslide area ($\widehat{A_{LS}}$), multiplied by the total number of non-landslide samples ($N_{NL}$):

5  (4)

$$N_{NLS} = \left(\frac{A_{LS}}{\widehat{A_{LS}}}\right) * N_{NL}$$

This correction allows us to resample our input data for completeness.  From each dataset, when then randomly selected 180 747 observations for analysis, as the number equal to the number of landslides in the smallest inventory (2014 Ludian

earthquake). In this way, data from each earthquake holds an equal weighting in the analysis.

An implicate assumption is that predictor variables are measured without error. Considering the known high uncertainty in ground motion, we sought to prevent our model from over-fitting. We apply a sample weighting based on the inverse of reported ground motion uncertainty when fitting our model (Pedregosa et al., 2011), such that the fit favours observations in

inverse proportion to the measurement error.

To develop a generalized model of hillslope failure probability, that has the greatest applicability to all earthquakes, we carried out our analysis by first fitting logistic regression models to data from each earthquake individually. From testing all possible combinations of predictor variables, we identified variable sets that exhibited a physically plausible and consistent

influence on $P_{LS}$. Whilst the regression coefficient value associated with a variable may differ between the two events, the latter condition stipulates that the direction of influence (indicated by a positive or negative coefficient value) must remain constant. Using these variable sets, we fitted combined logistic regression models using data from all 10 earthquakes. We then identified the variable set producing the highest levels of fit at two scales:

1.  Using the area under the receiver operating curve (AUC) method (Lee and Choi, 2004) we assess the fit of our
model at the finest (30 m) scale, to assess the ability to distinguish between landslide and non-landslide areas.
2.  We assess the fit of our model at the event-scale, by considering the accuracy of total landslide probability and landslide area predictions with respect to the associated error (residuals).

To reduce the likelihood of model over-fitting, we assess model fit using a test sample of 20% of observations held out of the model fitting process. We summarize our model building process in Figure 3.






### 3 Results

We find that a simple model combining PGA and hillslope gradient provides the most numerically elegant and best fitting model. The use of topographic variables other than hillslope gradient were found to produce models with a lower fit, or exhibit influences that were not ubiquitous across all earthquakes, either individually or in combination with hillslope

gradient. We find a similar result for ground motion variables. For instance, peak ground velocity (PGV), exhibits a positive correlation with failure probability across all earthquakes, but alone produces a lower goodness-of-fit than PGA. When PGA and PGV are modelled in combination, relationships are inconsistent across all earthquakes.

Using these predictor variables, we now identify the most appropriate model fit for landslide prediction.  In the case of all

earthquakes, $P_{LS}$ is positively correlated with both PGA and hillslope gradient, although the coefficient of the relationships varies (Figure 4). This is reflected in the combined logistic regression relationship, fitted using data from all 10 earthquakes (Figure 4, Combined model 1). However, this relationship appears to be problematic, as it predicts unrealistically high landslide probabilities for low gradient, low PGA locations. Correspondingly, the relationship underestimates landslide probability for high gradient, high PGA locations, relative to several of the relationships fitted for individual events. The

source of this issue is data from the Honshu, Iwate and Niigata earthquakes (Figure 4 – see relationships for individual events). Probability values for the 2011 Honshu earthquake (a deep subduction earthquake) are lower than for the shallow continental earthquakes in our dataset. The observation that subduction earthquakes trigger fewer landslides as compared to shallow continental events has been made previously (Wartman et al., 2013, Marc et al., 2016), attributed to a more significant attenuation of seismic waves. Our result suggests this is not explained by weaker peak ground accelerations

alone, as has been hypothesized (ibid.), or indeed by other variables included in our analysis. In the absence of data from additional subduction events to further investigate this effect, we exclude the 2011 Honshu data from our analysis from here on, and focus on a generalized fit for shallow, continental earthquakes only. For the Iwate and Niigata earthquakes, over-prediction of landslide probability for low PGA locations is likely the result of unconstrained spatial heterogeneity in ground motions across the areas affected by these events, resulting in an unrealistic relationship. It remains unclear whether there is

some further physical reason for the anomalous behaviour of earthquakes affecting Japan. However, in previous work we have shown how the effect of regional variations in lithology acts to modify the logistic regression coefficient assigned to slope gradient (Parker et al., 2015). This effect is broadly evident in both PGA and slope regression coefficients, when categorising earthquakes by their main lithologies. Volcanoclastic lithologies (present in the Japanese earthquakes) derive the    lowest    coefficients,    with    sedimentary/metasedimentary    lithologies    deriving    mid-range    coefficients,    and

metasedimentary/igneous lithologies deriving the highest coefficients.





By removing data from the Japanese earthquakes from our model fitting, we find that the alterative model better generalises observations from different regions, while providing a more physically plausible relationship (Figure 4, Combined model 2). For these reasons, we argue this is more appropriate to serve as a generalised 'global' model.

### 3.1 Model fit

We first assess the spatial fit of our model relative to the training data. We assess the distinction between landslide and non-landslide areas, using the area under the receiver operating curve (AUC) method (Lee and Choi, 2004). AUC values can be interpreted as the probability of a landslide observation being assigned a higher predicted probability than a non-landslide observation. For random guesses, AUC is equal to 0.5. By general rules of thumb, AUC values greater than 0.8 represent good classification performance (e.g.: Mehdi, 2011), while to be successful predictions must achieve an AUC value of 0.7

(Kritikos et al., 2015). AUC values for individual events range from 0.75 to 0.88 (Figure 5). In the case of Northridge, Chi-Chi, Wenchuan and Haiti, AUC values greater than 0.8 represent good classification performance, with Coalinga and Ludian AUC values of 0.75 and 0.78 just outside of this.

By aggregating landslide probabilities for each earthquake, it is possible to compare the total proportion of hillslopes with

landslide occurrence predicted and observed (Figure 6A). We find a positive correlation between observed and predicted probabilities ($R^2$ = 0.35), logit-transformed with respect to probability limits of 0 and 1. Predicted probabilities have a standard deviation of residuals of 0.65 logit-probability units. When these probabilities are applied across each study area, we find a good fit between observed and fitted total landslide area by event ($R^2$ = 0.83, Figure 6B). Correspondingly, predicted areas have a standard deviation of residuals of 0.27 orders of magnitude.

To estimate the uncertainty involved in using the model to predict landslides for earthquakes not included in the model training process, we set up a cross-validation experiment. We held data from individual earthquakes out of the training sample and used the fitted logistic regression coefficients to generate predicted probabilities for that earthquake. We then tested the accuracy of those predictions against the observed occurrence of landslides in each test event. This procedure was

repeated for each earthquake in turn, allowing us to estimate the out-of-sample predictive uncertainty and the model's ability to predict landslide activity for earthquakes with no mapped landslide data. For the Coalinga, Northridge, Chi-Chi, Wenchuan, Haiti and Ludian earthquakes, we fitted combined logistic regression coefficients to data from five events and predicted $P_{LS}$ for the sixth. As neither the Niigata or Iwate earthquakes are included in our training sample, we made predictions for these events based on the combined logistic regression model for the Coalinga, Northridge, Chi-Chi,

Wenchuan, Haiti and Ludian earthquakes.





The performance of the model in terms of distinguishing between landslide and non-landslide locations is the same as reported above for events included in model training. This is because the signs (positive or negative correlation) of relationships characterised by our model are ubiquitous for all earthquakes studied. As a result, relative spatial probabilities produced by the logit-linear relationships in the model are insensitive to changes in regression coefficients. Additionally,

predictions for the Niigata and Iwate earthquakes have lower AUC values of 0.73 and 0.62, respectively.

Uncertainty in prediction of absolute levels of landsliding for the test events is slightly higher in the case of out-of-sample predictions (Figure 7). This is to be expected, as the test events have no influence over the absolute predicted probability values. As the model under-predicts aggregate probabilities for the additional Niigata and Iwate earthquakes, which were not

included in training samples, this results in an overall tendency for predicted probabilities to be biased high by 0.34 logit-probability units, with a standard deviation of residuals of 0.82. This corresponds to a bias of 0.14 orders of magnitude in total landslide area with a 0.34 order of magnitude standard deviation of residuals. This represents a conservative (over-prediction) bias, in the case of our predicted outputs. Note that if this out-of-sample prediction test is applied to the 2011 Honshu off-shore subduction earthquake (using a model trained with data from the Coalinga, Northridge, Chi-Chi,

Wenchuan, Haiti and Ludian earthquakes), aggregate probability is over-predicted by over 4 logit-probability units, resulting in more than a 1.5 order of magnitude over-prediction in total landslide area.

**4 Discussion**

**4.1 Comparison with other earthquake-triggered landslide models**

Our model is designed to assign probabilities of landslide occurrence across the landscape, similar to Nowicki et al. (2014)

and Kritikos et al. (2015). Advantageously, our model provides absolute probability values, which are corrected for censoring of small landslides in the inventory to avoid under-estimating their net influence. This is important for geomorphological applications, where small landslides contribute a significant proportion of total areas and volumes of material (Malamud et al., 2004a). Similarly, for assessments of co-seismic landslide hazard, small landslides represent the majority of landslides and are still capable of significant impacts (Kargel et al., 2016). The absolute probability values our

model generates are also specific to hillslope failure, rather than the aggregate probability of failure and runout. This is important as procedures appropriate for analysing landslide runout are necessarily different from procedures for analysing landslide failure (e.g.: Clerici et al., 2010). Compared with previous studies, this factor is particularly evident at the relatively high (30 m) spatial resolution of our model, where most landslide source and runout zones can be separated. The failure-based probability output from our model may be combined with a predictive landslide runout model to more accurately

determine the net probabilistic footprint of landslides from these two constituents.



In seeking to provide a single model to describe a wide range of earthquakes, our approach is similar to that of Marc et al. (2016, 2017). Uniquely we have sought to leverage existing ground motion data, rather than to derive our model from more fundamental seismological principles. We argue that this is more appropriate to our approach, as we seek to capture regional variations in landslide probability, rather than event-level summary attributes, on the basis that spatial granularity is

important for the geomorphic and hazard consequences of co-seismic landsliding. In principle, the model proposed here, and that by Marc et al. (2017) should provide equivalent or comparable results, using total landslide area calculated from aggregate probabilities, or if we were to run simulations with our model to estimate likely total landslide distribution extent areas. Similarly, Robinson et al. (2016) have developed on the approach of Kritikos et al. (2015), to estimate numbers of landslides triggered by New Zealand earthquakes, by correlating spatial landslide frequencies from past earthquakes with a

fuzzy-logic-based susceptibility model. From this output, it is possible to estimate total landslide areas and volumes via Monte Carlo simulations using the landslide area- or volume-frequency distribution (Equation (1)), and scaling relationships between landslide area and volume (e.g.: Larsen et al., 2010). Hence, results from our approach and that of Robinson et al. should be equivalent when inverted using these models.

Our model is also less physically explicit than the model proposed by Gallen et al. (2015, 2016), in that we do not breakdown the probability of failure as governed by probable displacement and an assumed displacement threshold. By driving our conditional probability prediction directly with observations of hillslope failures from past events, we try to avoid the use of physical parameters, which may in some instances be poorly constrained. This is useful, as it allows us to generate predictions without material properties data, alongside an estimate of the likely uncertainty associated with that

prediction. Marc et al. (2016) handled this problem similarly, by applying a constant material sensitivity parameter and hypothesising a variety of mechanisms by which unconstrained material control may have influenced outlier events, while Gallen et al. (2016) dealt with this uncertainty by exploring a range of probable material properties. Through the alternative method presented here, our analysis derives similar outcomes; primarily that there is a general behaviour common to all the earthquakes tested here, and that departures from that general behaviour appear to relate to spatial, and perhaps temporal

(Brain et al., 2017; Parker et al., 2015), variations in material control on landscape response.

## 4.2 Further investigating sources of unconstrained variability

Our model performs well in predicting the spatial distribution and absolute probabilities of landslides, for shallow continental earthquakes. However, variability in model performance, particularly highlighted by the cases of the Niigata and

Iwate earthquakes, highlights the importance of further investigating sources of unconstrained variability in earthquake-triggered landslides. The analysis of departure from a general model is particularly pertinent when it comes to investigating this. Various hypotheses for this have been proposed, but remain to be tested alongside other spatial and temporal confounding factors, within a multi-variable generalised model. For instance, Dellow and Hancox (2006) proposed that New Zealand earthquakes occurring following wet periods have larger distribution areas than earthquakes occurring in dry



periods. Brain et al. (2017) proposed that low magnitude earthquakes drive the consolidation and stabilisation of hillslope materials, during periods intervening large earthquakes, while Parker et al. (2015) suggested that accrued brittle damage can leave unfailed hillslopes at greater susceptibility to failure following large earthquakes. Samia et al. (2017) have suggested that the occurrence of landslides destabilises hillslopes more widely, resulting in greater susceptibility for follow-up

landslides over a period of about 10 years. The events included here and in the growing global dataset span a range of likely conditions resulting from these spatial-temporal effects. Marc et al. (2016, 2017) have demonstrated the value of examining likely reasons why characteristics of earthquake-triggered landslide distributions outlie the global average, on an event-by-event basis, highlighting seismic fault characteristics, which influence the spatial distribution of seismic waves, and substrate properties, which influence hillslope susceptibility to landslide triggering. While considering these factors at the event-scale

provides a useful first-order indication of major sources of variability, each earthquake-triggered landslide event reflects the spatially complex interaction of these factors. Posing our analysis in terms of spatial landslide probability, has the potential to help disentangle consistent sources of variability in space and time (e.g.: Parker et al., 2015), and ultimately build new variables into the analysis to improve the success of predictive models.

Considering how the accuracy of model predictions break down with scale for different earthquakes and regions, may contribute to addressing this. Owing to the law of large numbers (e.g.: Bolthausen and Wüthrich, 2013), our model predictions of absolute probability naturally have lower uncertainty when aggregated at larger scales, while the prediction becomes increasingly uncertain when estimating landslide probability across smaller areas of interest. For further study, examining the breakdown of model uncertainty with scale may provide further indication as to the spatial wavelength of

unconstrained drivers of variability in landslide occurrence and thereby aid the identification of means to constrain these effects. A key challenge remains in finding adequate proxy variables to incorporate into our general model, with which to constrain sources of this variability and provide a basis for improving the success of the model.

## 5 Conclusions

Using logistic regression, we have developed a generalized model to describe and predict the spatial distribution of

earthquake-induced landslides. Our model predicts the absolute spatial probability of hillslope failure as a function of peak ground acceleration and hillslope gradient, based on data from global topographic and seismic ground motion datasets. The predicted output from our model provides a generalised or average landscape response, for landslides triggered in sedimentary, meta-sedimentary and igneous lithology, with a tendency to under-predict landslides in volcanic lithology, and applicable to shallow continental earthquakes within the moment magnitude range 6.2 to 7.9, and depth range 10 to 21 km.

Although our analysis tentatively suggests a systematic lithological effect on landslide probability, in the absence of consistent global lithological data, this remains to be tested further or constrained sufficiently to include in our model. Our model is notably simpler than other models developed through the spatial probability approach, by including only variables



that could be constrained consistently at the global-scale, and eliminating those that did not influence landslide probability in a consistent manner across all 9 earthquakes in our dataset. Using freely available topographic and ground motion datasets, our model can be applied to earthquakes with no landslide data, to provide first-order estimates of the probabilistic spatial distribution of landslides. Our predicted outputs may also provide a baseline to further investigate spatial and temporal

sources of unexplained variability in landslide distributions.

**6 Data availability**

Model predictions for applicable earthquakes are available online and open access at: shakeslide.geol.cf.ac.uk.

*Author contributions:*
RNP conducted the analysis and wrote the paper with input from NJR and TCH.

*Competing interests:*
The authors declare no conflicting interests with the work presented in this study.

*Acknowledgments:*
We thank the Natural Environment Research Council (NERC grants NE/J009067/1 and NE/N012240/1) and the Willis Research Network for funding to develop this work. We thank Girish Kumar and Ian Thomas for their help with web development, and Odin Marc and Siobhan Whadcoat for their assistance in contributing and helping compile landslide data,

as well as numerous authors cited who have made this work possible by making their data publically available. We also thank Alexander Densmore for his guidance, and David Petley and Nicholas Cox for their help developing earlier ideas that led to this work.

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



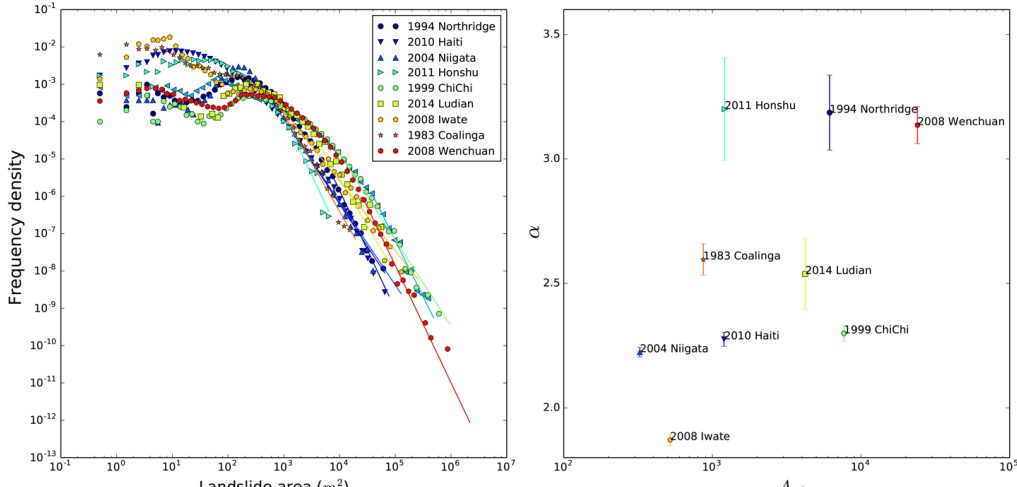

**Figure 1: (A) Landslide frequency density as a function of landslide source area for landslide inventories used in this investigation.**
**Data points represent the frequency-density (frequency divided by bin size calculated across logarithmically-spaced bins, after**
**Malamud et al. (2004a), Malamud et al. (2004b)), while the solid lines show the fitted power-law functions up to the points of**
**rollover initiation. (B) Best fit $A_{min}$ and $\propto$ (including $\sigma$ standard error) parameters for each landslide inventory.**



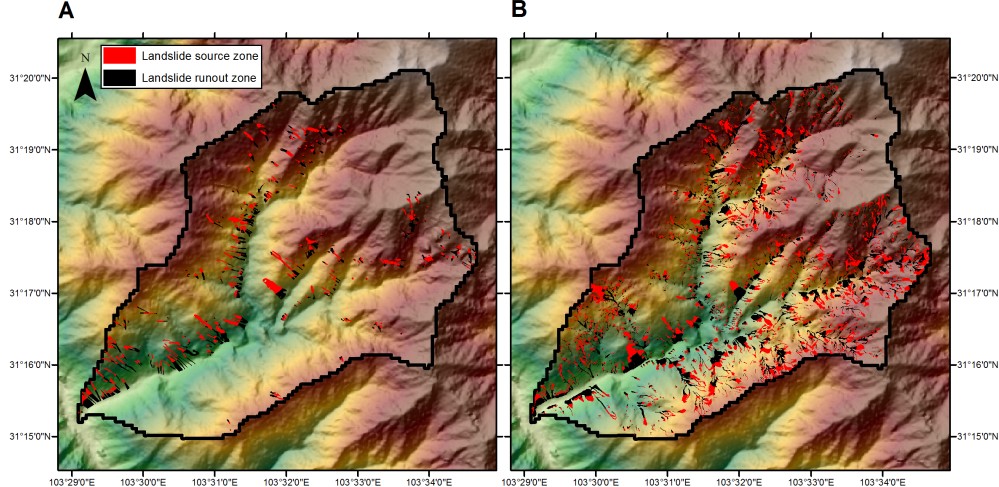

**Figure 2: Comparison of 2008 Wenchuan earthquake landslide inventory datasets. A) Li et al. 2014 landslide inventory mapped from 0.5-10m imagery. B) Sample high-resolution landslide inventory produced for this investigation mapped consistently from <0.5m resolution Digital Globe imagery provided by Google Earth.**





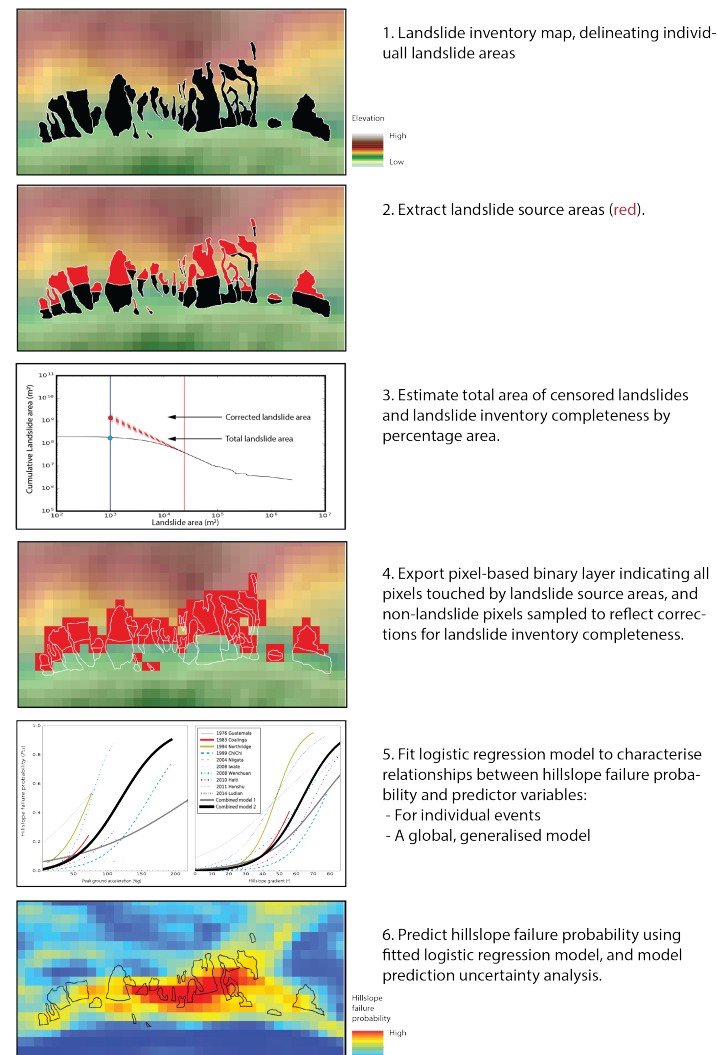

**Figure 3: Summary of model building process.**




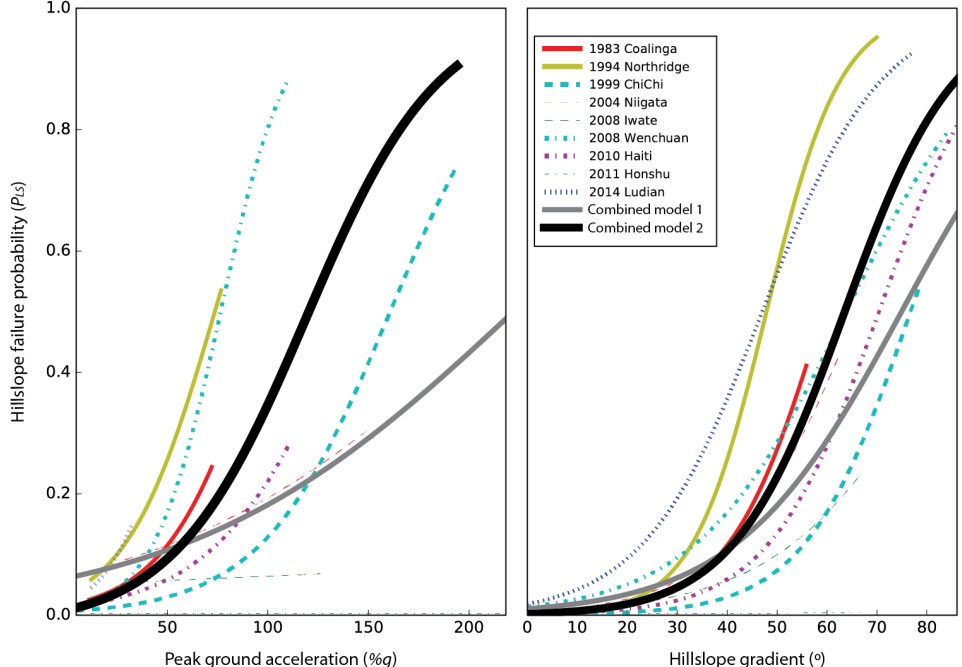

**Figure 4: Logistic regression relationships of landslide probability against PGA and hillslope gradient for individual earthquakes and our best fit model for all events. Relationships are displayed at typical values of predictor variables. PGA relationship is show for a hillslope gradient value of 40°. Hillslope gradient relationship is shown for a PGA value of 50 %g.**




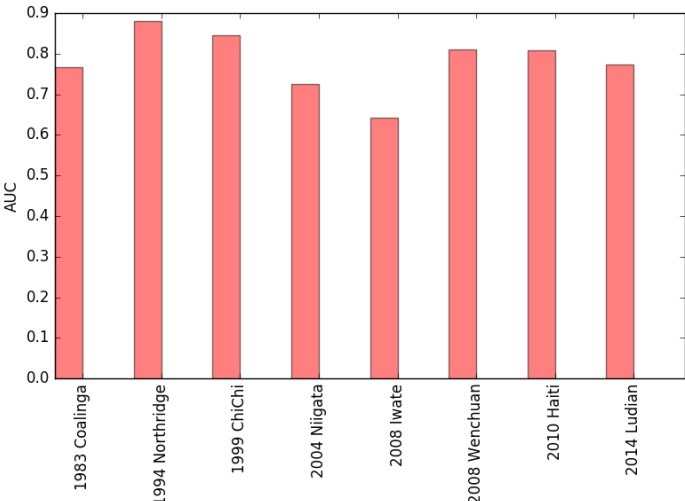

**Figure 5: Assessment of model ability to separate landslide and non-landslide areas. Receiver operating curve, area under the curve (AUC) values calculated for individual earthquake datasets.**

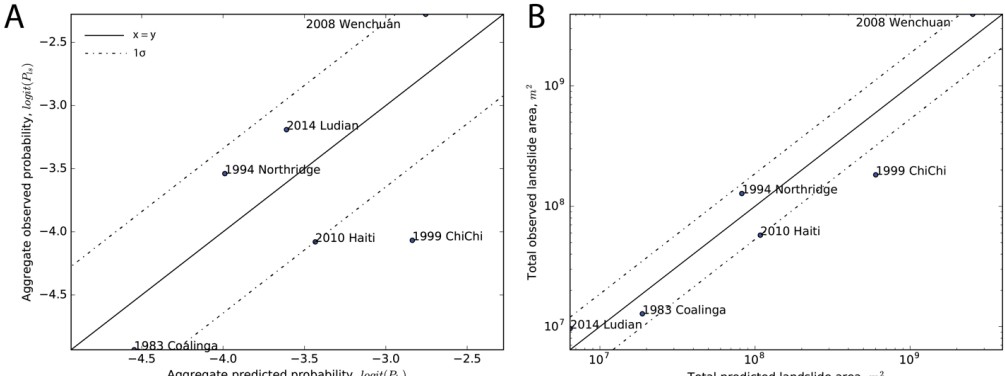

5 **Figure 6: Comparison of observed and fitted mean landslide probabilities (A) and total landslide areas (m²) (B) for individual earthquakes.**




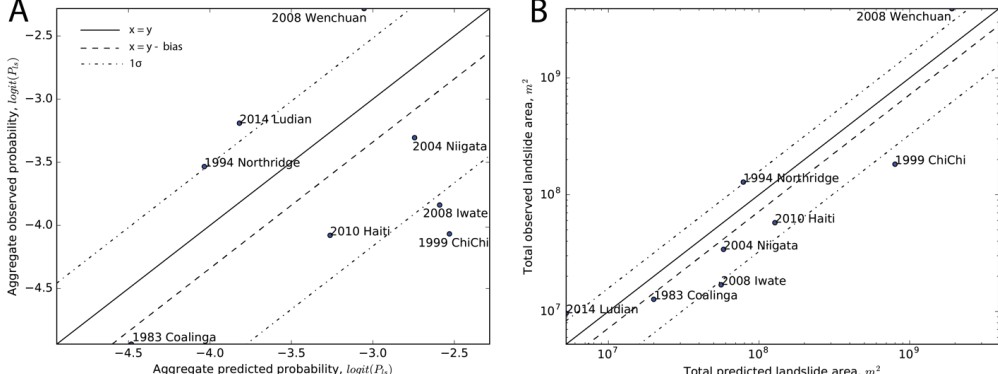

**Figure 7: Assessment of out-of-sample predictive accuracy, when using logistic regression model to predict landslide probability and area for earthquakes without landslide data. (A) Comparison of observed and predicted probabilities. (B) Comparison of observed and predicted landslides areas ($m^2$). Solid lines indicate x=y. Dashed lines indicate the conservative bias of predictions. Dash-dot lines indicate the standard deviation of residuals ($1\sigma$).**