# Peer review of "Spatial prediction of earthquake-induced landslide probability"

_Natural Hazards and Earth System Sciences, 2017_

## Referee Comment (RC1) · O. Marc (Referee) · 12 Jul 2017

Review comments for: Parker et al.,    Spatial prediction of earthquake-induced landslide probability

GENERAL COMMENTS
The paper by Parker et al., attempt at building and validating an empirical model of earthquake-induced landslide probability. The novelty and significance of the work comes from 3 important aspects:
1/ a very parsimonious model, only including essential widely available parameters : Shakemaps PGA and local slope gradient.
2/ the calibration and validation of the model with a relatively large number of earthquakes (9) in different context.
3/ A proposed method to obtain absolute probability and not only relative as is often the case in other studies (Nowicki et al. 2014, Kritikos et al., 2015)

The paper is interesting, clearly written and as the potential to make a good contribution in NHESS. However I highlight below some concern about the method to account for censored landslides and thus obtain absolute probability.
I also think the validation section could be a bit enlarged with some discussion about the spatial accuracy of the probability map.

SPECIFIC COMMENT :

I think the comparison of aggregated probability across the epicentral areas (defining a "predicted total area" ) to the actual estimate of total area is smart and allows to obtain absolute probability prediction from the model.
However, I think that there a two important problems in the methodology:

1/ the areas over which landslide probability must be integrated must be objectively definable *a-priori* , not based on the landsliding extent as it seems to be the case now.
If not the absolute probability could only be obtained after mapping and delimiting the extent of landsliding, which would be pointless.
Marc et al. 2016, 2017 may offer some tools for that, by giving a priori estimates of the extent of landsliding or of the total area. But the author may find a solution without this external model, as the PGA decay with distance is inherently included within the Shakemap and thus in the probability map. Setting to zero landsliding below an appropriate threshold of probability, or adjusting the relation between Pls and PGA may actually  result in reasonable estimate of At. I think the authors should explore that.

2/ As correctly noted by the author landslide catalogue may not always be complete, while an integral of the modelled probability should be compared to the exact total landslide area. Therefore they propose to use the frequency-size distribution to assess the completeness of the dataset and if needed estimate the amount of missing landslide area and correct the total area estimate. They estimate the missing proportion due to censoring to be up to ~60% or ~90% for the Northridge and Wenchuan inventories. My objections are summarized here and developed in details in the Line by Line comments.

I think these results are erroneous.
The test catchment they used to validate it in the Wenchuan area is unconvincing for several reasons:
 - Many missing slides are large (i.e., not missing in the original catalogue because of censoring)

- The imaged used to map at finer scale is taken 3 years after the earthquakes and likely includes rainfall induced landslides. The number of post-seismic landslides may even be very significant if unfailed slopes were damaged by the earthquake.

For Northridge, mapped with air photos and field survey, (Harp and Jibson 1996 states that they could map landslides until size of a few meters, meaning there resolution limit is probably around 100m² not 1000m². They also state: "From our field investigations, we estimate that we missed no more than about 20% of the landslides that exceeded 5 m")

The author estimates that the dataset is 40% complete. Thus total landslide area should be more than 20km², meaning several thousands of landslide of 1000-5000m² (i.e. dimensions 30-70m long for square landslides). The author do not discuss where could be all these missing landslides.

I think a more plausible solution to these problems is that the censoring is over-estimated. Mainly because the authors suppose the catalogues of Wenchuan, Northridge ChiChi, should have a power-law decay continuous until the minimum size Amin observed in other inventories (Niigata, Iwate). That is more or less equivalent to assume a similar roll-over position and that the transition from roll-over happens in the same way in each inventory.

I think that 1/ there may be several physical reasons explaining, why some cases may have a roll-over at larger area (say 500m instead of 50m ). 2/ the transition at the roll-over may be very sharp (e.g., Niigata, figure 1) or very broad ( Haiti figure 1), while both cases are considered complete and well mapped. I would say a broader transition may be due to the merging of different lithology/domain with different surface properties (and thus different rollover position)

In summary, I think the censoring estimation methods should absolutely be re-evaluated before publication. My personal view is that for all the catalogue considered censoring is always below 10-30% of At and define an uncertainty range. The problem for Wenchuan is that the Li et al., 2013 catalogue is incomplete because it misses visibility in some areas, but I would recommend requesting the most complete catalogue (that has about 3-4 times more landslides) and using it, rather than trying to correct for censoring.

This will change the corrected estimates of total area used to validate the model.

Then finding an *a-priori* way to obtain total area and absolute probability would make the paper of greater interest..

Finally, I think including a discussion about the agreement between the spatial probability pattern and the landslide pattern (and not only the bulk pattern) would strengthen the paper (see detailed comment). Indeed if this is not done the probability map may have a reasonable aggregated probability for the wrong reasons (i.e. with local probability uncorrelated to the landslide pattern.)

In any case, although what I suggest represent some additional analysis, I think it is within reach of the authors and that it would make the paper more rigorous and impacting.

I am waiting to see the refined analysis and remain available for further discussion if needed.

LINE BY LINE COMMENTS:
Good Introduction.

P6 Line 1-4: Ok. But the authors may want to include the recent Roback 2017 dataset (downloadable there https://www.sciencebase.gov/catalog/item/582c74fbe4b04d580bd377e8  )
This dataset also separated source from full slides. Thus it may be an interesting independent test of their scar estimation techniques. Further it may be an interesting case to add in their empirical

probability model.

P 6 : Line 10 : amalgamation : What about blurring effect ? The fact that low resolution imagery do not only cause the omission of small landslides but may cause multiple neighbor landslides to be confused with a single large one, when the imagery cannot resolve undisturbed areas in between. See Fig 2d,e in Marc and Hovius 2015. I think this should be mentioned here as this is an important caveat.

Table 1 : Nice summary table. Cite Oglesby et al., 2000 maybe for hanging wall effect. And Meunier et al., 2013 for the lithology differences.

Table 2 : Why to give, strike dip rake if not discussed ?

Table 3 : Specify what you mean by "edited by Marc et al 2016" ?
And there must be some confusion because in the publication by Marc et al., 2016 , they used Gorum et al., 2013 for the Haiti earthquake and did not access any catalogue for the Ludian and Coalinga earthquake. Then they stated that Northridge, ChiChi (and other cases not used here) were edited (spli manually) above a certain size. And that Niigata and Iwate did not seem to have significant amalgamation.

Table 3 : How did you consider liquefaction slides within the Wartman dataset ? Did you remove them ? I guess yes as they report ~ 3400 "slides" including 1000 lateral spread.
I think it is correct to remove these spread/liquefaction events as they are different from the others (especially in terms of slope) but it should be indicated somewhere.

Table 3: completeness above 10^3 km² ... I guess you mean m² ? Please correct.

Wenchuan case: check At = 396 km²... From Xu 2014 catalogue we have ~200,000 slides and 1100km², that is ~4 times more slides and 3 times more area.
 Then I wonder if the use of the Li et al., 2014, catalogue makes sense, while I would suggest to request the dataset from Xu et al. 2014 and test your model with it.

Next, it ask the question of the accuracy of the censoring correction. 13% completeness, then it suggests that At should be ~ 4000 km² ? It makes no sense, the Xu catalogue is already very exhaustive, and mapping landslides in a greater area of interest.
 Actually, by superimposing Xu mapping on top of SPOT-5 imagery from just after the quake, I can see that there is often some over-estimation of landslide area, because of mapping artifacts caused by low-resolution imagery in some areas (blurring, see previous comment). In any case it must certainly be close of an upper bound for your censoring correction.

P9-10 : How do we know that it is resolution censoring ? Could be physical differences changing the position of the rollover.

P10 Line 4-8: The authors assume that the power-law behavior must be conserved until a universal size Amin=1000m², equivalent to say that all roll-over should be below this size.
However, Stark and Guzzetti 2009 (and also Katz and Aharanov 2006) propose (based on theoretical and experimental modeling, respectively) that the position of the roll-over is fixed by the transition from a weak surficial layer to higher cohesion bedrock. Then, a deeper transition would create a roll-over at larger size. For example, Stark and Guzzetti 2009 have a model where the rollover is ~500m for a depth of transition to bedrock of 2.2m And because landslide area scales almost as the square of

landslide depth (Guzzetti et al., 2009, Larsen et al., 2010), a regolith extending in average to 4 or 6 meters would give roll-over at ~ 2000m² or ~4500m².

P10 Line 14:  The censoring corrected At is not visible in Table 3.
To be sure it should be At_corrected = At / (1-Completeness), correct ? Seems to be the case on Figure 6.
IMPORTANT: You refer to Appendix 1 and 3 (not 2 ?) But none or them are in the main text or supplement. It would be nice to see them through publication in the open discussion. If this relates to supplementary figures, be more specific.

P10 Line 16: Again I do not believe the censoring estimate is correct. Northridge was carefully mapped with high resolution airphotos taken around the areas. Although the detailed mapping is sometimes problematic (i.e. amalgamation) I don't see where can be the missing 60% of the landslide zones.

P10 Line 23-25 and Fig 2.
There are many issues with Fig 2 and the associated "test" of the censoring correction in the Wenchuan case.

In Fig 2 you need to show the original imagery used in each case. If not all at least some zoomed part where we can see the disturbed areas. It can be a supplementary figure but it is important to see why landslides were missed or mapped differently.

There are also strange things in the left panel, with large landslides crossing ridges. Possibly elongated in north-west / south-east direction (possibly due to poor orthorectification of the original imagery). Also what is the resolution of the image used at this site, 0.5, 1 or 10 m ? It is importqnt to assess mapping quality.
Further, I would like to see the CDF of landslide area in the right and left panel. It is clear that a large number of additional small landslides are mapped, possibly because of higher resolution. But a large number of medium to large slides are missing on the left panels, and this can not be resolution censoring. Is it clouds ? If yes is this area representative in terms of clouds cover compare to the rest of the mapping zone ?

You did not indicate the date of the google earth image used to map the left panel landslides.
 From Google Earth I see that there is a pre image in 2005 and then the next image is in 2011 !! This is thre years after the earthquake, and there is a significant chance that a good proportion of the landslides in 2011 are due to following rainfall. We even expect higher rate of rainfall induced landslides with unfailed slopes likely damaged (Marc et al. 2015). Thus I do not think this example really support the claim that the inventory is 13% complete.

P12- Line 5: About Shakemaps for PGA.
What do you mean topographically based site amplifications ? This is a very ambiguous wording, because it suggest topographic amplifications (eg., Meunier 2008) is accounted for in ShakeMaps whereas I am quite sure they are not.
 I guess you refer to (Allen and Wald 2009) that use topography (actually smoothed slope gradient) as a proxy for the presence of thick soil/sediment cover (in flat areas) or bedrock (in steep areas) to derive an estimate of Vs30 and therefore of a potential site amplification due to a slow velocity shallow layer. This is different from the resonance of topographic ridge considered to impact landsliding.
Exactly for this reason the fact that Shakemap use a relatively large radius of influence to adjust the

strong motion to the closest station may be a bias, because stations are almost always in alluvial plains, with possibly a quite different site effects than from the ridge1km away where landslide may occur.

P12 Line 16 : Yes. Citing Korup 2008 would also be relevant there, for the link between lithology and slope gradient PDF.

P13 Line 6 : I don't understand this sentence. I think there are typos there. Plus, what is 180747 refering to ? Table 2 says there are ~1000 landslide in the Ludian dataset.

P13 Line 20: Is Pls the landslide density (area/area of the pixel) or a number density ? Not sure it was defined before, please do it here or at an earlier point.
On figure 4 you say it is hillslope failure probability. Still this may be clarified better in the text, especially in the context of the resolution of your analysis, 30m, that is quite small.

P14 line 2: "numerically elegant" ? Do you mean less computationally intensive ? Or most parsimonious ? Or possibly both ?

P14 Line 5 : Yes, PGV and PGA values are very correlated in Shakemap. Maybe too much.
Anyway, how much "less good" is PGV fit compared to PGA ? Is it really significantly less good or likely within the error of any of these 2 variables or of the model uncertainty. This is important for future worker to decide on testing both or only one of the two variables.

P15 Line 1-3 : If your generalized global model excludes all volcanic areas from its fit it seems to me that it is not global anymore, and should be described as a general model for sedimentary/metasedimentary and igneous rocks.
Then why not have also a "volcanoclastic model " based on the 3 Japanese cases ?
Anyway, I see that you summarized that better later in the conclusions, and wording may be adjusted in the main text.

P15 Line 5-10 : Clearly there are many ways to discuss success of a prediction. I am not very familiar with the advantages of using AUC. I see that it allow you to avoid to have to decide on a threshold to say if a cell has a landslide or not. However, how to interpret a 0.05 probability within a 30m pixel ? Given that many actual landslides are as large or larger than your pixel size it cannot be seen as a proportion of the pixel affected by landslides. Further AUC score is a bulk score that does not help much to evaluate the spatial validity of your prediction.

 I think it would make sense to expand a bit this validation discussion. Sure it is nice to compare the integrated probability to the average aggregated landslide density, but all the point to make a spatial prediction (as a probability map) is to compare it quantitatively to the landslide spatial pattern.

Maybe showing a the proportion of landslide scar falling in a series of probability bin ? And the proportion of non landslide cell in probability bin ? Something like Fig 14 of Jibson et al., 2000.
 I also think you could maybe show the effect of averaging your probability at, say, 90m and 300m to show how the results are improved, even if exact location of landslides becomes less accurate.

Finally, by engaging themselve to define Ldsl / Non Ldsl pixel with a probability threshold the author could also evaluate False Positive / Fals Negative. See for example Nowicki et al., 2014 Fig 6 a.

This may make the model easier to compare to the rest of the literature.
In such graph we can see the trade off between correct prediction and false positive, as well as unpredicted landslide can be seen easily for various threshold.

P16 Line 23 : "This is important for geomorphological applications, where small landslides contribute a significant proportion of total areas and volumes of material (Malamud et al., 2004a)."

I disagree stongly with this sentence and don't think that Malamud et al., 2004a said that. In a relatively complete catalogue it is easy to show that the total  volume  is dominated by large landslides (see Marc et al., 2016).
 For total area it is a bit more variable and it depends what you call small. For Iwate, I calculated 67% of total area is due to slides larger than $10^4$ m². For Northridge and Niigata this slide category accounts for ~25% of the total area. But 87%  or 67% of the total area is due to slides $>10^3$ m² for these two cases, respectively.

In contrast, the next sentence about hazard is fairer as small slides are order of magnitude more frequent, and can still be a significant hazard. Still even if rare large landslide (by area or volume) even if rare can create a cascade of hydrosedimentary hazard (through landslide dam break, river aggradation or avulsion or simply debris flow) that can pose risk to entire valley and multiple settlement. So it is not only about frequency of given slide size ranges.

P16 Line 25: The author recall that there model describe probability of failure (that is of landslide scar). However I do not think they quantified how different the results are when taking the scar only or the full source. This could be a nice and simple test to perform.

P16-P17: The whole section 4 could be a bit more quantitative: How is the accuracy of your model compared to Nowick or Kritikos's ? This is especially relevant as you examined the same cases: Chi Chi, Northridge, Wenchuan. Probably, your model will be less good, but it has only 2 parameters and is more global. So it would be nice to have some comparison.

P17 Line 1-5:  True increasing the spatial resolution of  EQIL prediction is an important challenge. But in this sense I think you should engage further the local comparison of you probability map to the landslide map.

P17 Line 25 Cite Marc et al., 2015 together with Brain and Parker as it deals with the same topics with contrasting results.

Conclusions: Clear and Fair summary. Again it would be nice to see some quantitative accuracy measure here. Allowing to compare such global, parsimonious prediction to more parameter-rich prediction, adjusted on 1 or 2 cases only (Nowicki 2014, Kritikos 2015).

P21: Marc et al., 2017  is now out of review and published so please update the ref :
 Marc, O., P. Meunier, and N. Hovius (2017), Prediction of the area affected by earthquake-induced landsliding based on seismological parameters, *Nat. Hazards Earth Syst. Sci.*, *17*(7), 1159–1175, doi:10.5194/nhess-17-1159-2017.

P21 MEHDI, T. 2011. Kernel smoothing for ROC curve and estimation for thyroid stimulating hormone. International Journal of Public Health Research, 239-242.

I wonder if this paper is general enough to support your statement about "good value" of AUC.
I would think there are more work in the geoscience/natural hazard/remote sensinc communities using AUC that can be cited instead.

Figure 4 : Some of the dashed line are difficult to see. Please make them thicker.

Figure 6: I think 6A would be more informative if showing the aggregated Pls (in log scale if necessary), rather than the logit (that you can't convert intuitively).
In 6B and 7B you you do not plot observed total area, but the one corrected for censoring. This should be indicated in the caption or axis legend.

**References used in the review not present in the manuscript:**

Allen, T. and D.J. Wald (2009b). On the use of high-resolution topographic data as a proxy for seismic site conditions (VS30), *Bull. Seism. Soc. Am.* 99(2A), 935-943.

Gorum, T., C. J. van Westen, O. Korup, M. van der Meijde, X. Fan, and F. D. van der Meer (2013), Complex rupture mechanism and topography control symmetry of mass-wasting pattern, 2010 Haiti earthquake, *Geomorphology, 184*, 127–138, doi:10.1016/j.geomorph.2012.11.027.

Jibson, R. W., E. L. Harp, and J. A. Michael (2000), A method for producing digital probabilistic seismic landslide hazard maps, *Engineering Geology*, *58*(3–4), 271–289, doi:10.1016/S0013-7952(00)00039-9.

Korup, O. (2008), Rock type leaves topographic signature in landslide-dominated mountain ranges, *Geophysical Research Letters*, *35*(11), n/a–n/a, doi:10.1029/2008GL034157.

Katz, O., and E. Aharonov (2006), Landslides in vibrating sand box: What controls types of slope failure and frequency magnitude relations?, *Earth and Planetary Science Letters*, *247*(3–4), 280–294, doi:10.1016/j.epsl.2006.05.009.

Oglesby, D. D., R. J. Archuleta, and S. B. Nielsen (2000), The Three-Dimensional Dynamics of Dipping Faults, *Bulletin of the Seismological Society of America*, *90*(3), 616–628, doi:10.1785/0119990113.

Roback, K., M. K. Clark, A. J. West, D. Zekkos, G. Li, S. F. Gallen, D. Chamlagain, and J. W. Godt (n.d.), The size, distribution, and mobility of landslides caused by the 2015 Mw7.8 Gorkha earthquake, Nepal, *Geomorphology*, doi:10.1016/j.geomorph.2017.01.030.

Xu, C., X. Xu, X. Yao, and F. Dai (2014), Three (nearly) complete inventories of landslides triggered by the May 12, 2008 Wenchuan Mw 7.9 earthquake of China and their spatial distribution statistical analysis, *Landslides*, *11*(3), 441–461, doi:10.1007/s10346-013-0404-6.

---

## Referee Comment (RC2) · PhD Gallen (Referee) · 7 Aug 2017

**Review of Parker et al., "Spatial predictions of earthquake-induced landslide probability"**

**Summary of research:**

Parker et al. use logistic regression analysis of 9 coseismic landslide inventories from different locations and combine these results with a Monte-Carlo routine to estimate absolute spatial probabilities and probability uncertainties of earthquake triggered landsliding for shallow continental earthquakes. Through the initial regression analysis the authors demonstrate that slope and peak ground accelerations (PGA) yield the simplest and most generally applicable fits to the global data set and note that subduction zone events appear to behave differently than other events. The model is tested using out-of-sample data from each of the nine landslide inventories used in the study and preforms reasonably well given its simplicity. The value of the study and the newly proposed model is that it provides a baseline to study unexplained variability in coseismic landslide distributions (e.g. rainfall patterns, structural geology, material strength, etc.) and it allows for rapid assessment of coseismic landslide hazards quickly after an earthquake using freely available DEMs and USGS Shakemap products.

**General reviewer assessment:**

This manuscript is clear, well-written and should be published in NHESS. The methods used are sound and the model derived will be useful for coseismic landslide hazard assessment due to its simplicity. I was happy to see the authors use multiple landslide inventories for the regression analysis and work to find the most relevant and easily implemented parameters (slope and PGA) into a hazard model. The summary of previous research is thorough and even handed, and in most aspects the discussion of the results is detailed and exhaustive. Furthermore, Parker et al. do not try to over sell this new and useful model, but clearly state weaknesses, offer meaningful suggestions for future research, and how explain how their model might be used to address other questions in natural hazards and geomorphology.

I have three "major" comments that I would like the authors to address. I put major in quotations because I think that these comments can be easily addressed by the authors, and thus recommend this manuscript be accepted pending minor revisions.

(1) The Wenchuan earthquake comprises nearly half of the landslides used in this study based on the inventory by Li et al., 2014 (Note: that a more complete inventory by Xu et al., 2014 does exists with nearly 200,000 mapped landslide, but perhaps the authors had difficulty getting access to this data set). I am curious as to how much the Wenchuan event effects the model output. In other words, if this event were excluded from the analysis, how different would the results be? At the very least I would like to see a discussion on the potential influence of Wenchuan on the combine model results because I it might have a strong effect on final results.

(2) I would like to see an assessment of the performance of the model in a discrete spatial sense. The benefit of being able to predict spatial probabilities is that it will provide insight on the factors that contribute to the spatial distribution of landsliding, but it is not entirely clear how well this model preformed in a spatial sense. The figures showing how well the model preformed in event-by-event are good, but a figure demonstrating more quantitatively (e.g. beyond the spatial comparisons shown in the supplement) how well the model preformed spatially for a given event would be great.

(3) The discussion on the influence of material strength is generally lacking in the final section in the discussion "further investigating sources of unconstrained variability". At least from theoretical considerations, material strength is as important to seismic slope stability as topographic slope, while

peak ground accelerations play a secondary role. I understand the difficulty in constraining material strength parameters that are relevant to landsliding on regional scales, which is why Parker et al. and many other research choose to ignore material strength. The manuscript does note that rock strength is likely accounted for in some way with the regression analysis, and that based on comparisons of different events lithology seems to play a role. However, I am curious if the approach employed in this study might be exploited to better understand the spatial patterns of material strength for a single event, even if in a relative sense. It would be nice to see some discussion on this point as it is important for both natural hazards and landscape evolution and I think it would broaden the discussion.

I also have one suggestion that I think could help elevate the profile of the manuscript: test the model against the Gorkha (Nepal) earthquake, which is not used in this study. My colleagues and I produced a landslide inventory for the Gorkha earthquake that includes GIS polygon files of source areas and full landslide extents that is freely available from the USGS (Roback et al., 2017a,b; https://www.sciencebase.gov/catalog/item/582c74fbe4b04d580bd377e8). Such a comparison would help to demonstrate the applicability of the model presented in this study. However, such a comparison is only a recommendation that the authors may choose to ignore.

Some minor comments and technical corrections are offered below.

Please let me know if you have any questions.

      Kind regards,

      -    Sean Gallen

**Specific reviewer comments (line-by-line):**

p.2, l.25-28: qualify what is meant by censoring here. This usually has to do with the resolution of the imagery used to map (e.g. resolution bias).

P2-3., l.33-34, 1-2: I appreciate the citations here, but I'm not sure that this characterizes our work accurately. I would cite our 2016 paper in the final sentence of the paragraph, as it was sort of a negative results paper that emphasizes the importance of material strength on the slope stability equations we used. Our 2015 work uses these physically-based models in conjunction with topographic slope from DEMs, estimates of PGA from ShakeMap, and a coseismic landslide inventory to invert for material strength parameters at large spatial scales. To validate the results we compare a forward model using calculated material strength to the observed landslide distribution, so I guess this reference loosely fits in with the second-to-last sentence in the paragraph.

P.4: some additional references for the case study column: Seismic wave attributes, Godt et al., 2008; Orientation of hillslope relative to seismic source, Harp et al. 2014; Bedrock lithology, Dreyfus et al., 2013.

p.5, l.15: why is McFadden et al., 2005 cited here? This study has to do with physical weathering of boulders and cobbles due to diurnal temperature changes.

p.10, l.18-25: I find this discussion and rationalization confusing. Is this simply to justify the low levels of completeness of some inventories?

P9-10, l.1-30, 1-25: After reading this section, it is not clear to me how estimates of completeness are used. It isn't until page 13 that this information is divulged. I would suggest adding a few sentences in this section describing how these estimates of (in)completeness are relevant and will be used in subsequent modeling.

p.12, l.18-33: Is the model derived in this study only applicable to SRTM 1-ArcSecond (~30m) data? I would assume that this is the case because it is calibrated to slope derived from a DEM of that resolution and slope is highly dependent on DEM resolution (e.g. Larsen et al., 2014). Please state explicitly that this model is calibrated to a specific DEM resolution and that it should not be applied using DEMs of different resolution.

**Technical corrections (line-by-line):**

p.5, l. 31: space between elevation and (Parker et al., 2015).

p.6, l.3: add a period to the end of the sentence.

p.10, l.12-13: I do not understand what is being said in this last sentence. Please clarify.

p.13, l.1: 9 instead of 10?

p.13, l.8-10: I am struggling with this sentence. "we" instead of "when"? Is this number 180,747 or 180 or 747? How does this number related to the Laudian earthquake with 1024 landslides?

p.13, l.14: Pedregosa et al., 2011 is not in the references.

p.17, l.8: "built" instead of "developed"?

References:

Correct the journal titles to the Copernicus format.

p.25, l.14: journal, volume, page numbers?

p.26, l.36: volume, page numbers?

p.26, l.40: was this published in 2008 or 2009?

p.26, l.49: earthquaker in journal title.

p.27, l.56: volume?

p.28, l.11: page numbers.

p.29, l.10: check journal title.

**References cited in review that are not in the manuscript:**

Dreyfus, D., Rathje, E. M., and Jibson, R. W., 2013, The influence of different simplified sliding-block models and input parameters on regional predictions of seismic landslides triggered by the Northridge earthquake: Engineering Geology, v. 163, p. 41-54.

Harp, E. L., Hartzell, S. H., Jibson, R. W., Ramirez-Guzman, L., and Schmitt, R. G., 2014, Relation of Landslides Triggered by the Kiholo Bay Earthquake to Modeled Ground Motion: Bulletin of the Seismological Society of America, v. 104, no. 5, p. 2529-2540.

Larsen, I. J., Montgomery, D. R., and Greenberg, H. M., 2014, The contribution of mountains to global denudation: Geology, v. 42, no. 6, p. 527-530.

Roback, K., Clark, M. K., West, A. J., Zekkos, D., Li, G., Gallen, S. F., Chamlagain, D., and Godt, J. W., 2017a, The size, distribution, and mobility of landslides caused by the 2015 Mw7.8 Gorkha earthquake, Nepal: Geomorphology.

Roback, Kevin, Clark, M.K., West, A.J., Zekkos, Dimitrios, Li, Gen, Gallen, S.F., Champlain, Deepak, and Godt, J.W., 2017b, Map data of landslides triggered by the 25 April 2015 Mw 7.8 Gorkha, Nepal earthquake: U.S. Geological Survey data release, https://doi.org/10.5066/F7DZ06F9.

Xu, C., Xu, X., Yao, X., and Dai, F., 2014, Three (nearly) complete inventories of landslides triggered by the May 12, 2008 Wenchuan Mw 7.9 earthquake of China and their spatial distribution statistical analysis: Landslides, v. 11, no. 3, p. 441-461.